# Generalized Anxiety among Swiss Health Professions and Non-Health Professions Students: An Open Cohort Study over 14 Months in the COVID-19 Pandemic

**DOI:** 10.3390/ijerph182010833

**Published:** 2021-10-15

**Authors:** Thomas Volken, Annina Zysset, Simone Amendola, Agnes von Wyl, Julia Dratva

**Affiliations:** 1Institute of Health Sciences, ZHAW Zurich University of Applied Sciences, 8401 Winterthur, Switzerland; annina.zysset@zhaw.ch (A.Z.); julia.dratva@zhaw.ch (J.D.); 2Faculty of Medicine and Psychology, Sapienza University of Rome, 00185 Rome, Italy; simone.amendola@uniroma1.it; 3School of Applied Psychology, ZHAW Zurich University of Applied Sciences, 8005 Zurich, Switzerland; agnes.vonwyl@zhaw.ch; 4Medical Faculty, University of Basel, 4056 Basel, Switzerland

**Keywords:** generalized anxiety disorder, health professions, students, COVID-19, mental health, young adults

## Abstract

To date, little is known about the long-term trajectory of generalized anxiety disorder (GAD) symptoms in health professions (HP) students over the course of the pandemic. Like health professionals in general, HP students may have a significantly greater susceptibility to GAD symptoms due to their involvement in the health care system and the associated specific stressors and risks during the COVID-19 pandemic. The HEalth in Students during the Corona pandemic study (HES-C) provided the opportunity to investigate the long-term course of GAD symptoms with eight measurement points over 14 months in 9380 HP and non-HP students in Switzerland between March 2020 and June 2021. We employed logistic regression models with clustered sandwich standard errors to estimate unadjusted and adjusted prevalence of GAD symptoms. In the full model, we adjusted for age, gender, nationality, social status, social support, self-efficacy, and COVID-19 symptoms in the past 4 weeks. At baseline, the estimated adjusted GAD symptom prevalence was 17.6% (95% CI = 14.4–20.7) in HP students and 24.4% (95% CI = 22.3–26.5) in their peers. With the peak of the second SARS-CoV-2 infection wave in October/November 2020, GAD symptom prevalence substantially increased and then remained stable over time, despite changes in the epidemiological situation and its associated containment measures. At the last follow-up in June 2021, GAD symptom prevalence in HP and non-HP students was 22.9% (95% CI = 16.3–29.5) and 36.9% (95% CI = 32.9–40.9), respectively. Absolute differences in GAD symptom prevalence between student groups over all eight measurement points ranged from 6.2% to 14.9% (all *p* < 0.05). Non-HP students are identified as a specifically vulnerable group. Accordingly, target group-specific public health campaigns and interventions should be developed with the aim to strengthen their resources, reducing GAD symptoms, and preventing chronification.

## 1. Introduction

The rapid and global spread of SARS-CoV-2 infections since the early 2020s has posed fundamental, existential challenges to governments, scientists, and the public. In their efforts to contain the spread of the virus, governments have drastically restricted civil liberties, imposing measures, such as curfews, quarantines, assembly bans, and mandatory home offices. Since the first wave of SARS-CoV-2 infection reached Switzerland in March/April 2020, the country has been hit by a second and third wave in September/October 2020 and February/March 2021, and there are strong indications that Switzerland might be faced with a fourth wave in September/October 2021. Overall, people have been confronted over a long period with constantly changing epidemiological threats, restrictions, and unpredictability. Stressful events represent important risk factors for the emergence of anxiety symptoms and difficulties in regulating negative emotions [1,2,3]. Moreover, feelings of loneliness, intolerance of uncertainty, worry, and fear generalization are related to symptoms of anxiety [4,5,6,7].

The consequences of containment measures, such as social isolation, boredom, and financial problems, have considerably increased mental distress in adults in comparable situations [8,9]. Jeong and colleagues found increased levels of psychological distress, e.g., anxiety and anger, during isolation for Middle East respiratory syndrome [10] and similar findings have been reported during the outbreak of novel swine-origin influenza A [11]. Moreover, anxiety levels closely mirrored the daily number of new cases during the 2003 outbreak of severe acute respiratory syndrome in Hong Kong [12]. Recent studies showed that substantial proportions of the general population reported moderate-to-severe symptoms of anxiety during the initial outbreak of COVID-19 in Iran (28%) [13], China (29%) [14], the United Kingdom (24%) [15], Canada (20%) [16], Belgium (15%) [16], Switzerland (12%) [16], the Philippines (23%) [16], and Bangladesh (37%) [17]. The percentages of moderate-to-severe anxiety in China and the UK were lower before the COVID-19 pandemic [18,19]. A substantially lower prevalence of anxiety symptoms before the pandemic as compared to the pandemic has also been reported for the United States of America (17.9%) [20] and the German adult population (44.9%) [21]. Hetkamp and colleagues found that six weeks after the lockdown, anxiety levels in the general population of Germany remained elevated [22]. However, in the United Kingdom and Switzerland, a decrease in anxiety levels has been observed in the general population 20 weeks after the lockdown [15] and in university students near the end of the first infection wave after 6 weeks [23], suggesting a habituation to the threatening situation. In contrast, Généreux et al. found that the prevalence of anxiety symptoms in the Swiss adult population remained stable between May and November 2020 [24]. Furthermore, several recent studies raise concerns for the mental health of younger people [24,25,26,27,28,29], particularly university students [30,31,32,33], who appear to be more vulnerable to depression and anxiety symptoms. Similarly, healthcare workers [25,27,34,35,36,37,38,39] were at high risk of mental illness. A recent systematic review and meta-analysis on generalized anxiety disorder among health care workers during the pandemic found that the prevalence for this specific occupational group was 32.0% and identified anxiety disorders as one of the fundamental psychological problems [40]. However, evidence seems to be mixed. The prevalence of anxiety disorder symptoms was 14.9% in a national sample of internal medicine physicians in the United States of America [41], i.e., much lower than reported in previous systematic reviews [35].

To date, little is known about the long-term trajectory of generalized anxiety disorder (GAD) symptoms in health professions (HP) students over the course of the pandemic. Like health professionals in general, HP students may have a significantly greater susceptibility to GAD symptoms due to their involvement in the health care system and the associated specific stressors and risks during the COVID-19 pandemic. Moreover, the existing evidence among the few studies that assessed changes in anxiety levels in the course of the pandemic is mixed, relate only to changes over a few weeks or months, and the majority are available only for the general adult population. In the present study, we therefore aimed to (1) examine the course of GAD symptoms in HP and non-HP students over 14 months of the COVID-19 pandemic, (2) to compare levels of GAD symptoms among HP students and their peers, and (3) to identify factors associated with GAD symptoms. Monitoring and understanding different levels of GAD symptom susceptibility and its associated factors in students is critical for differentiated evidence-based public health interventions that are appropriate for the target group. The HEalth in Students during the Corona pandemic (HES-C) study provided the opportunity to study the long-term course of GAD symptoms with eight measurement points over 14 months in this under-investigated population.

## 2. Materials and Methods

### 2.1. Survey Design and Study Design

Study data stem from the HES-C study, which aims to (1) evaluate the health of students during the pandemic, (2) investigate changes in health behavior and associated factors, as well as (3) assess students’ perception of the pandemic and related measures and their impact on students’ lives. All enrolled students at the Zurich University of Applied Sciences (ZHAW) (N = 13,500) were invited to participate in nine consecutive surveys that were administered between April 2020 and June 2021 in an open cohort design. Open cohort studies allow subjects to enter and exit the study population throughout the entire study, thus better reflecting the realities of universities. Each survey lasted about 20–25 min and ran for a total period of seven working days. Participants’ informed consent was obtained before each survey. Methods and questionnaire construction have been published in detail by Dratva and colleagues [42]. In the present study, we used the pooled data from eight of the nine survey waves (N = 9380). Data collection of the included waves took place from 3 April 2020–14 April 2020, 30 April 2020–11 May 2020, 5 October 2020–13 October 2020, 30 November 2020–10 December 2020, 21 January 2020–29 January 2021, 8 March 2021–16 March 2021, 3 May 2021–11 May 2021, and 14 June 2021–23 June 2021. The measurement time points covered different epidemiologic situations and governmental constraints to contain the pandemic (Figure 1). The study was approved by both the local cantonal ethics committee (BASEC-Nr. Req-2020-00366) and the ZHAW data protection officer.

### 2.2. Measures

#### 2.2.1. Outcome: Generalized Anxiety Disorder Symptoms

We assessed generalized anxiety disorder symptoms using the Generalized Anxiety Disorder scale (GAD-7) [43], which measures self-reported anxiety levels of participants in the past 2 weeks. The GAD-7 includes seven items to be rated on a 4-point Likert scale, ranging from 0 (not at all) to 3 (ever day). Items are summed, resulting in a total score ranging from 0 to 21, with higher scores indicating higher levels of anxiety. Based on the total score, 4 levels of anxiety severity were derived: minimal (0–4), mild (5–9), moderate (10–14), and severe (5–21) [43]. As a screening instrument for GAD symptoms, a GAD-7 score of 10 or greater has been proposed, yielding a sensitivity of 89% and a specificity of 82% for GAD [43]. We used the latter threshold score to classify students as having generalized anxiety disorder symptoms (0 = no, 1 = yes).

#### 2.2.2. Predictor and Covariates

For our primary predictor, participating students were classified into those studying in the Department of Health Professions and those studying in other departments (0 = non-health professions students, 1 = health professions students). Non-health professions students included students from the following faculties: Applied Linguistics, Applied Psychology, Architecture, Design and Civil Engineering, Life Science, Facility Management, School of Engineering, School of Management and Law, and Social Work.

Sociodemographic covariates included gender (0 = women, 1 = men), age at last birthday in complete years centered at the mean value, nationality (0 = Swiss, 1 = foreign), and mean-centered social status of parents at student age 16 using a modified McArthur scale ranging from 1 (lowest) to 10 (highest) [44].

Psychosocial covariates included mean-centered self-efficacy and social support. We used the Allgemeine Selbstwirksamkeit Kurzskala (ASKU; General Self-Efficacy Short Scale) [45] to assess self-efficacy. The ASKU comprises three items (“In difficult situations, I can rely on my abilities”, “I can cope with most problems on my own”, and “Generally, I can handle strenuous and complicated tasks well”) and participants were asked to respond on a five-point Likert scale (from 1 = “strongly disagree” to 5 = “strongly agree”). The ASKU scale scores range ranges from 1 to 5 and represents the mean score over all three items, with higher scores indicating higher self-efficacy. Social support was assessed using the Oslo Social Support Scale (OSSS-3) [46]. The OSSS-3 explores social support through three items about the number of close confidants, sense of concern or interest from other people, and relationship to neighbors. We used the established cut-off values of the OSSS-3 sum scores [46] to classify respondents into individuals with strong (0 = 12–14), poor (1 = 3–8), and moderate (2 = 9–11) social support.

COVID-19-related symptoms were assessed using the following statement with binary response option (0 = no, 1 = yes): “Have you had symptoms in the past 4 weeks that would be compatible with a COVID-19 infection? For example, cough (usually dry), sore throat, shortness of breath, and fever, muscle pain”.

In order to capture time trends, covariates for time (0 = baseline measurement at M1, 1 = [M2, M3,…M8], and interactions between the latter and our primary predictor were used.

### 2.3. Statistical Analyses

Descriptive statistics (i.e., frequencies, percent, mean, and standard deviation) were applied to evaluate the characteristics of the samples. We used one-way ANOVA and Chi-square-tests to assess mean-level stability between measurement points. Logistic regression models with a clustered sandwich estimator for standard errors, which adjust for repeated measures of the same subject, were used to estimate adjusted generalized anxiety disorder symptoms, i.e., the conditional probability of generalized anxiety disorder symptoms given the model predictors (Equation (1)):(1)$$\mathrm{Pr}\left(y_{j}\neq 0|x_{j}\right)=\frac{exp\left(x_{j}\beta \right)}{1+exp\left(x_{j}\beta \right)}$$
where *x*_j_ refers to the model predictors 1 through *j* and *β* refers to the respective unknown parameter to be estimated.

Models were gradually adjusted. Starting with a restricted model (Model 1) with only the primary predictor and time variables, subsequent models introduced additional covariates for demographics (Model 2: age, gender), social origin (Model 3: parents’ social status, nationality), and psychosocial covariates (Model 4: self-efficacy, social support, COVID-19 symptoms). The purpose of this procedure was to evaluate the stability of the primary predictor over the course of the pandemic, since not all covariates were recorded at all measurement time points. For Models 1 and 2, all involved variables were available at all time points (M1–M8). Model 3 comprised 5 measurement time points (M1, M3, M6–M8), and Model 4 comprised 4 measurement time points (M1, M3, M6, M8). For all logistic models, we visually checked whether independent variables were related linearly to the log odds, checked an adequate minimum of 10 cases with the least frequent outcome for each independent variable, and assessed multicollinearity using the variance inflation factor (VIF) in the multivariable models. Underlying assumptions of the logistic models were met in all models. We report odds ratios (ORs) with corresponding 95% confidence intervals (95% CIs), predictive margins (average predicted probability), and average marginal effects. Statistical significance was established at *p* < 0.05. We used Stata Version 15.1 (StataCorp, College Station, TX, USA) for statistical analyses.

## 3. Results

### 3.1. Participants’ Characteristics

Over all eight included survey waves, 9380 students participated in the study, and 70.6% were women and 29.4% were men (Table 1). Mean age was 25.8 years, and 91.7% were Swiss nationals. The percentage of health professional students was 23.8%, which is considerably higher than their share in the total student population (13.0%). Mean parents’ social status was 5.6, which is almost exactly the middle of the scale and means that most students located themselves in the social middle class.

With respect to psychosocial covariates, overall mean self-efficacy was high (3.8), and most students could count on at least medium (59.7%) or even high (22.5%) social support. However, 17.8% of students reported low social support. Finally, 16.4% of the students said that they experienced COVID-19 symptoms in the past 4 weeks before the survey.

With the exception of parents’ social status, all statistical tests for mean-level stability between measurements indicated statistically significant differences. However, due to the large number of observations and the associated statistical power, this should not be overestimated. In particular, a look at the range shows that the values do not vary strongly between the surveys. For example, the proportion of health professions students ranges from 20.8% to 30.7%, the proportion of women ranges from 68.6% to 74.9%, and self-efficacy ranges from 3.7–3.8 with standard deviations between 0.6 and 0.7.

### 3.2. Generalized Anxiety Disorder Symptoms: Trajectory, Risk, and Protective Factors

Table 2 shows the GAD-7 scores and generalized anxiety disorder levels according to the established threshold scores [47]. Across all waves, the mean GAD-7 score was 7.0, and the proportion of students with moderate and severe anxiety scores amounted to 17.4% and 9.0%, respectively. Hence, the crude prevalence of generalized anxiety disorder symptoms was 26.4%. With respect to mean-level stability, GAD-7 scores, the proportion of participants in GAD-7 categories, and the proportion of students with GAD symptoms varied substantially between survey waves (all with *p* < 0.001). From baseline measurement (M1) to the first follow-up (M2), the prevalence of anxiety symptoms first declined, then increased substantially at the third (M2- > M3) and fourth (M3- > M4) follow-up, and remained thereafter in a range between 30.7% and 33.8%.

In a first restricted model (Table 3, Model 1), we estimated the effect of our primary predictor (faculty) and included contrasts for time (measurement points) and their interaction with faculty. The main effect of faculty (OR = 0.64; 95% CI = 0.50–0.82) indicated that health profession (HP) students as compared to those enrolled at other faculties were less likely to have GAD symptoms. Furthermore, the time contrasts revealed that non-HP students were significantly less likely to report GAD symptoms at the first follow-up at M2 (OR = 0.53; 95% CI = 0.52–0.75) as compared to their baseline GAD, were equally likely to report GAD symptoms at the second follow-up M3 as compared to their baseline GAD (*p* = 0.541), and from the fourth follow-up (M4) on, non-HP students were always substantially more likely to report GAD symptoms as compared to baseline. Interaction terms between time and faculty were statistically not significant, indicating that HP students share the same trajectory of GAD symptoms over time that we found for non-HP students, i.e., HP students also experienced a decrease in GAD symptoms at the first follow-up at M2 as compared to their baseline GAD, and from the fourth follow-up (M4) on were always more likely to report GAD symptoms as compared to baseline. Further adjusting for gender and age (Table 3, Model 2), parents’ social status and nationality (Table 3, Model 3), self-efficacy, social support, and COVID-19 symptoms (Table 3, model 4), we consistently found HP students as compared to their peers were less likely to report GAD symptoms (Model 2: OR = 0.57; 95% CI = 0.44–0.73; Model 3: OR = 0.59; 95% CI = 0.45–0.77; Model 4: OR = 0.67; 95% CI = 0.50–0.89). Similarly, time contrasts and interaction terms showed the same pattern as described for Model 1 above. With respect to the covariates, men as compared to women were less likely to report GAD symptoms. In the full model (Model 4, adjusting for all covariates), the corresponding OR was 0.70 (95% CI = 0.58–0.85). Older age was negatively associated with GAD symptoms in Models 2 and 3 but was no longer statistically significant in the full model. Furthermore, students from higher social status families were consistently less likely to report GAD symptoms (Model 4: OR = 0.93; 95% CI = 0.88–0.98) and nationality was consistently not significantly associated with GAD symptoms. With regard to psychosocial covariates, students with high self-efficacy were less likely (OR = 0.33; 95% CI = 0.29–0.37) and students with low social support as compared to those with high social support (OR = 2.36; 95% CI = 1.82–3.05) were more likely to report GAD symptoms. Finally, students who experienced COVID-19 symptoms in the past four weeks before the survey were more likely to report GAD symptoms (OR = 1.44, 95% CI = 1.17–1.77).

Figure 2 shows the trajectory of the estimated absolute GAD symptoms prevalence for HP students and their peers as well as the respective absolute difference over time, based on the restricted model (panel A, C) and the full model (panel B, D). In the restricted model, the estimated adjusted probabilities of reporting GAD symptoms at baseline, i.e., the proportion of students reporting symptoms, was 17.3% (95% CI = 14.1–20.5) for HP students and 24.5% (95% CI = 22.4–26.6) for non-HP students. At the first follow-up (M2), the prevalence of GAD symptoms had declined for both student groups (HP students: 8.8%; 95% CI = 5.7–11.8; non-HP students: 16.9%; 95% CI = 14.6–19.2). However, at the second follow-up (M3), GAD prevalence had increased to baseline levels again (HP students: 18.4%; 95% CI = 13.3–23.5; non-HP students: 24.1%; 95% CI = 21.3–27.0) and further increased between the second and third follow-up (HP students: 24.1%; 95% CI = 18.8–29.4; non-HP students: 32.7%; 95% CI = 29.5–35.8). Thereafter, GAD symptoms prevalence remained at the high level of the third follow-up. At the last follow-up, the estimated prevalence for HP students was 22.8% (95% CI = 16.1–29.6) and 36.6% (95% CI = 32.4–40.1) for non-HP students. Absolute differences in GAD symptoms prevalence ranged from 5.8% to 13.8%, with substantially lower prevalence in HP students. With the exception of the third and sixth measurement, where differences were only borderline significant (M3: *p* = 0.053, M6: *p* = 0.090), differences were all significant at *p* < 0.01. The full model yielded similar results. At baseline, the estimated GAD symptoms prevalence was 17.6% (95% CI = 14.4–20.7) in HP students and 24.4% (95% CI = 22.3–26.5) in their peers. At the last follow-up, prevalence had increased for both HP and non-HP students, and was 22.9% (95% CI = 16.3–29.5) and 36.9% (95% CI = 32.9–40.9), respectively. Absolute differences in GAD symptoms prevalence between student groups ranged from 6.2% to 14.9% and were all statistically significant.

## 4. Discussion

The crude prevalence of GAD symptoms in our student population in the March to September 2020 time frame (M1-M3) was very similar to that of the adult population in Switzerland, which was 22.3% [24]. After October 2020, the prevalence of GAD symptoms in the student population was substantially higher and remained high, ranging from 30.7% to 33.8% (M4-M8). While the change in GAD symptoms between April and November 2020 may well be associated with changes in the epidemiological situation, GAD symptoms prevalence remained at a level roughly equivalent to the peak of the second wave (M4), despite significantly lower numbers of new SARS-CoV-2 infections and related deaths in the following months. These findings are partly in line with previous studies that found that the evolution of the epidemiological situation, i.e., mortality and death rates, did not seem to influence GAD symptoms [24]. However, in our student population, this was only the case between January and June 2021, while between March and November 2020, we first observed a short-term decrease in GAD symptoms during the first infection wave, similar to a study in the United Kingdom [15], and subsequently an increase in GAD symptom prevalence that seemed to closely mirror the epidemiological situation. The missing coherent association between GAD symptoms and the latter, however, lends little support for the hypothesis suggesting a habituation to the threatening situation [23]. Moreover, unlike the GAD symptoms in the general adult population in Switzerland, which were stable between May and November 2020 [24], our student population experienced a substantial increase. These findings suggest that students were more susceptible to GAD symptoms as compared to the general population, which corresponds with previous study findings [30,31,32,33], and that their vulnerability increased over the course of the pandemic. It might be that students’ continued uncertainty about the course of the pandemic cumulated with uncertainty about continuing and completing their studies, leading to increased anxiety symptoms. In addition, students may have been more isolated and challenged by the rapid change to online distance learning than adults in the general population who performed their normal work in the home office [48,49]. Furthermore, previous findings indicated that symptoms of anxiety increased in students who were more concerned about their own health [23]. Consequently, the observed increased levels of GAD symptom prevalence over the course of the pandemic in our student population may be related to increased health concerns as well.

While both HP and non-HP students experienced increases in GAD symptom prevalence, the difference with respect to the general adult population in Switzerland was almost entirely due to non-HP students. Their crude GAD symptom prevalence was 32.6–37.0% while HP students were at a level comparable to the general population at 22.8–26.6%. The substantial higher GAD symptom prevalence of non-HP students as compared to the general population can most likely be attributed to their significantly lower average age. Previous studies found that younger age was associated with a higher risk of mental illness [24,25,26,27,29]. Similarly, age was negatively associated with GAD symptoms in our restricted models. This raises the question of why, despite their similar ages, HP students exhibit significantly fewer GAD symptoms than their peers, and exhibit fewer GAD symptoms than might have been expected based on their age alone. Several factors may have contributed to this. HP students are generally likely to have better health literacy than non-HP students in general and mental health literacy in particular [50]. This includes knowledge about sources of reliable health information but also the ability to better classify and understand this information based on their prior professional knowledge [51,52]. In addition, because of their involvement in the health care system, HP students often have first-hand experience or first-hand information and have professional contacts in the health care system. So far, however, the presumed relationships are merely hypotheses that should be investigated in future studies.

In all our multivariable logistic regression models, we found substantial and robust negative associations between HP students and GAD symptoms after adjusting for age, gender, social status, nationality, self-efficacy, social support, and COVID-19 symptoms in the past four weeks before the survey. In the full model, adjusting for all covariates (Table 3, Model 4), age was not associated with GAD symptoms while previous studies reported that younger people were at a higher risk of GAD symptoms [24,25,26,27,29]. However, most of the latter studies assessed populations with a much wider age range as compared to our student population. Hence the lack of variation in students’ age could be the reason why we could not detect a corresponding effect. Men as compared to women were less likely to report GAD symptoms, a finding that is corroborated by previous studies [17,21,24,33,53]. Students’ nationality was not associated with GAD symptoms. This finding is in line with previous studies from France [54]. However, the evidence seems to be mixed. In the United States of America, foreign nationality has been found to be positively associated with anxiety [55]. Moreover, we found low social status students to be more likely to report GAD symptoms. However, evidence from previous studies seems again to be mixed: low social status was associated with an increased likelihood to report GAD symptoms in China [56], and a study of Chinese students in Japan found that higher social status was associated with higher levels of anxiety [57].

With respect to psychosocial covariates, students with poor social support were more likely and students with higher levels of self-efficacy were less likely to report GAD symptoms, which is in line with previous French and Iranian studies assessing social support [54] and self-efficacy [58], respectively. Our results suggest that social support and self-efficacy are important protective factors in being able to cope with anxiety.

Finally, we found that having COVID-19 symptoms was associated with GAD symptoms. Previous studies reported similar results in French and Chinese university students [33,54,57].

The findings of our study should be interpreted keeping their limitations in mind. Firstly, we used self-reported measures to assess anxiety, which may have induced social desirability response bias. Secondly, more women participated in the study, which could have resulted in higher crude GAD symptom prevalence because generally women tend to report higher GAD scores [17,21,24,33,53]. In our adjusted models, however, we adjusted for gender. Thirdly, our student sample is not representative for young adults in general since they belong to a more highly educated population. Whether or not higher education proves to be a protective factor for the development of GAD symptoms is difficult to judge because prior population-based studies have demonstrated both positive [59], negative [60], or no educational effects [61]. Fourthly, 75–97% of students in our open cohort participated in only one survey, depending on the model and the number of measurement time points included in the model. Accordingly, our study has more the characteristics of a cross-sectional study with repeated measures and accordingly no causal inferences can be drawn. Fifthly, the different foci of the surveys and considerations of research economics meant that not all variables were collected at all survey time points. As a result, not all model parameters over all models could be consistently estimated using the same sample. While for our primary predictor, i.e., HP versus non-HP students and its trajectory over time, there were no substantial differences between the models in terms of either effect size or statistical inference, for some covariates, such differences did occur between the models. For the latter, in particular, it is thus not possible to conclusively determine whether these differences in effect size or statistical inference are original or due to the specific composition of the sample. Sixthly, previous studies found that subjects with chronic diseases, especially autoimmune diseases, were at particular risk for mental health disorders [62,63], and that the incidence of anxiety disorder may depend on subjects’ pre-existing health conditions rather than on their profession [63]. Due to the lack of data on autoimmune diseases, we were unable to control for this possible confounder. Future studies should systematically collect data and incorporate respective variables into their models.

## 5. Conclusions

The prevalence of GAD symptoms is high and has increased over time. This suggests that there were few habituation effects to the uncertain situational dynamics during the pandemic. HP students were significantly less affected by GAD symptoms as compared to their non-HP student peers. For students affected by anxiety symptoms, campaigns and interventions should be developed with the aim to strengthen their resources, and prevent chronification. Future studies should specifically examine the link between health literacy, access to trusted sources of information, and GAD symptoms in order to derive targeted programs to strengthen these skills, especially in non-HP students, if necessary.

## Figures and Tables

**Figure 1 ijerph-18-10833-f001:**
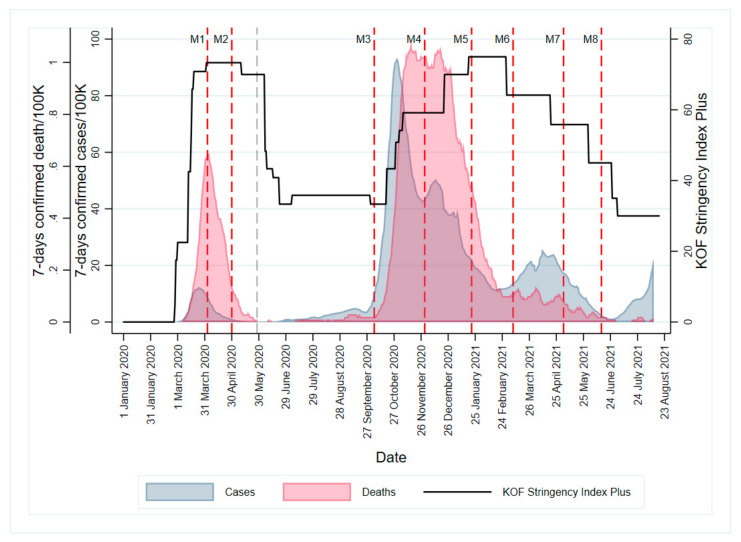
HES-C measurement points and the course of the COVID-19 pandemic in Switzerland. Legend: Vertical dashed lines indicate the start of each survey wave, with red lines referring to those included in the present study; 7-day confirmed cases and death per 100,000 population are daily moving averages over 7 days. The KOF (Konjunkturforschungsstelle) Stringency Index Plus is an adopted version of the Oxford Stringency Index, which allows assessment of national and cantonal measures. The index ranges from 0 (=no governmental containment measures) to 100 (=full lockdown).

**Figure 2 ijerph-18-10833-f002:**
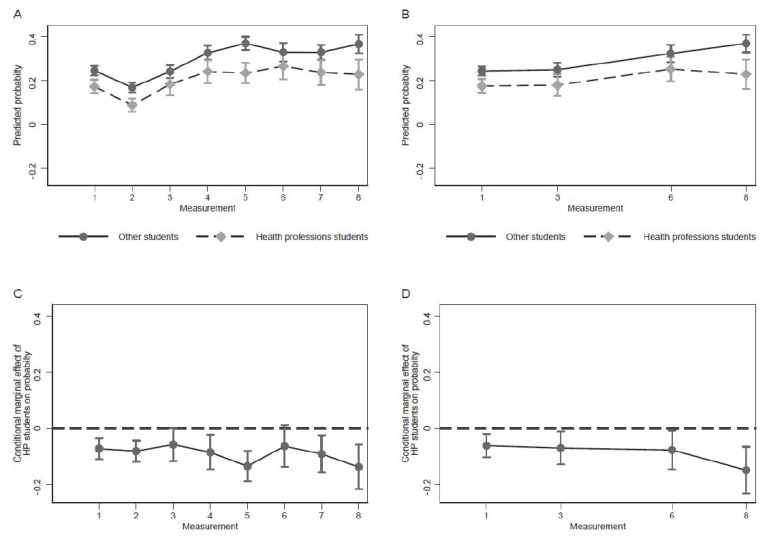
Predicted probability and conditional marginal effects of HP and non-HP faculty membership over time. Legend: Predicted adjusted probability of GAD symptoms for HP students and non-HP students in the restricted model (Model 1, panel (**A**)) and the full model (Model 4, panel (**B**)), covariates fixed at means. Panels (**C**) (restricted model) and (**D**) (full model) show respective differences in the probability of reporting GAD symptoms between HP students and non-HP students. Dots represent point estimates; whiskers represent corresponding 95% confidence intervals.

**Table 1 ijerph-18-10833-t001:** Participant characteristics.

	Measurement									
Variable	M1	M2	M3	M4	M5	M6	M7	M8	Total	*p*
Faculty (%)										<0.001
HP students	24.5	23.9	20.8	22.6	22.7	30.7	24.4	22.4	23.8	
Non-HP students	75.5	76.1	79.2	77.4	77.3	69.3	75.6	77.6	76.2	
Gender (%)										<0.05
Women	69.8	70.0	69.0	71.9	68.6	74.9	72.9	71.9	70.6	
Men	30.2	30.0	31.0	28.1	31.4	25.1	27.2	28.1	29.4	
Age (m ± sd)	26.4 ± 5.6	26.7 ± 5.8	25.0 ± 5.4	25.4 ± 6.0	25.4 ± 5.7	25.2 ± 5.8	25.3 ± 5.8	25.4 ± 5.5	25.8 ± 5.7	<0.001
Parents’ social status (m ± sd)	5.6 ± 1.6		5.6 ± 1.7			5.6 ± 1.7	5.7 ± 1.6	5.7 ± 1.7	5.7 ± 1.7	0.878
Nationality (%)										<0.05
Foreign	8.7		10.6			6.4	7.7	6.8	8.3	
Swiss	91.3		89.4			93.6	92.3	93.2	91.7	
Self-efficacy (m ± sd)	3.8 ± 0.6		3.8 ± 0.6			3.7 ± 0.7		3.8 ± 0.6	3.8 ± 0.6	<0.001
Social support (%)										<0.001
Low (3–8)	15.3		19.5			21.9		19.4	17.8	
Medium (9–11)	60.0		60.0			59.0		58.6	59.7	
High (12–14)	24.7		20.5			19.1		22.0	22.5	
COVID-19 symptoms (%)										<0.001
No	78.6		86.0		86.3			93.1	83.6	
Yes	21.4		14.0		13.7			6.9	16.4	
Number of observations	2170	1376	1079	1095	1376	700	926	658	9380	

*p* = probability (Pearson Chi-squared test for categorical variables; one-way ANOVA for continuous variables); m ± sd = mean ± standard deviation; HP = health professionals; Mx = measurement point.

**Table 2 ijerph-18-10833-t002:** Generalized anxiety disorder scores and levels over time.

	Measurement									
Variable	M1	M2	M3	M4	M5	M6	M7	M8	Total	*p*
GAD-7 score (m ± sd)	6.5 ± 4.4	5.6 ± 4.3	6.8 ± 4.5	7.6 ± 4.9	8.0 ± 5.0	7.8 ± 5.0	7.4 ± 5.1	8.0 ± 5.2	7.0 ± 4.8	<0.001
GAD-7 categories (%)										<0.001
Minimal (0–4)	38.7	48.3	35.5	31.5	27.2	31.6	33.6	30.3	35.6	
Mild (5–9)	38.5	36.8	41.4	37.7	39.0	37.1	35.8	36.2	38.0	
Moderate (10–14)	16.4	9.6	16.2	19.7	22.4	18.9	19.4	20.1	17.4	
Severe (15–21)	6.4	5.3	6.9	11.1	11.4	12.4	11.2	13.5	9.0	
GAD symptoms (%)										<0.001
GAD-7 score ≥ 10	22.8	14.9	23.1	30.8	33.8	31.3	30.7	33.6	26.4	

*p* = probability (Pearson Chi-squared test for categorical variables; one-way ANOVA for continuous variables); m ± sd = mean ± standard deviation; GAD = Generalized Anxiety Disorder; Mx = measurement point.

**Table 3 ijerph-18-10833-t003:** Logistic regression models: generalized anxiety disorder symptoms and its associations.

Variable	Model 1			Model 2			Model 3			Model 4		
	OR	*p*	95% CI	OR	*p*	95% CI	OR	*p*	95% CI	OR	*p*	95% CI
HP (Ref = non-HP)	0.64	0.001	0.50, 0.82	0.57	0.000	0.44, 0.73	0.59	0.000	0.45, 0.77	0.67	0.006	0.50, 0.89
Time (Ref = 1)												
2	0.63	0.000	0.52, 0.75	0.63	0.000	0.52, 0.75						
3	0.98	0.828	0.81, 1.19	0.94	0.541	0.78, 1.14	1.00	0.984	0.80, 1.24	1.01	0.922	0.80, 1.27
4	1.49	0.000	1.24, 1.79	1.45	0.000	1.21, 1.74						
5	1.80	0.000	1.52, 2.13	1.77	0.000	1.50, 2.09						
6	1.51	0.000	1.21, 1.88	1.47	0.001	1.17, 1.83	1.44	0.002	1.14, 1.82	1.35	0.023	1.04, 1.74
7	1.51	0.000	1.24, 1.83	1.46	0.000	1.20, 1.78	1.50	0.000	1.23, 1.83			
8	1.78	0.000	1.44, 2.20	1.73	0.000	1.40, 2.14	1.80	0.000	1.44, 2.24	1.96	0.000	1.54, 2.50
Time × HP (Ref = 1 × HP)												
2 × HP	0.74	0.166	0.48, 1.13	0.74	0.172	0.48, 1.14						
3 × HP	1.10	0.673	0.70, 1.73	1.14	0.582	0.72, 1.78	1.02	0.945	0.62, 1.68	0.94	0.809	0.57, 1.55
4 × HP	1.02	0.928	0.68, 1.53	1.03	0.870	0.69, 1.56						
5 × HP	0.81	0.281	0.56, 1.19	0.83	0.331	0.57, 1.21						
6 × HP	1.15	0.537	0.74, 1.78	1.15	0.526	0.74, 1.79	1.08	0.748	0.68,1.72	0.98	0.919	0.60, 1.59
7 × HP	0.99	0.957	0.64, 1.52	1.01	0.981	0.65, 1.54	0.96	0.845	0.62,1.49			
8 × HP	0.80	0.363	0.49, 1.30	0.81	0.403	0.50, 1.33	0.79	0.350	0.47, 1.30	0.71	0.208	0.41, 1.21
Men (Ref = women)				0.74	0.000	0.66, 0.84	0.74	0.000	0.63, 0.86	0.70	0.000	0.58, 0.85
Age				0.98	0.001	0.97, 0.99	0.98	0.004	0.97, 0.99	1.01	0.432	0.99, 1.02
Parents’ social status							0.89	0.000	0.85, 0.93	0.93	0.004	0.88, 0.98
Foreign nationality (Ref = Swiss)							1.19	0.156	0.94, 1.50	1.19	0.216	0.90, 1.57
Self-efficacy										0.33	0.000	0.29, 0.37
Social support (Ref = high)												
Low (3–8)										2.36	0.000	1.82, 3.05
Moderate (9–11)										1.15	0.197	0.93, 1.41
COVID-19 symptoms (Ref = no)										1.44	0.000	1.17, 1.77
Constant	0.33	0.000	0.29, 0.36	0.37	0.000	0.33, 0.42	0.35	0.000	0.31, 0.41	0.23	0.000	0.19, 0.29
Number of observations		9330			9329			4936			4022	
Number of subjects		6975			6974			4491			3891	
Pseudo R-squared (%)		2.6			3.0			2.5			12.20	

Logistic regression models with clustered sandwich estimator for standard errors; OR = Odds Ratio; *p* = probability; 95% CI = 95% confidence interval; HP = health professions students; Ref = reference category; generalized anxiety disorder symptoms refer to a GAD-7 score ≥ 10.

## Data Availability

The study data presented in this study are available on Zenodo data repository (https://zenodo.org/, accessed on 14 October 2021).

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
