# Peer review of "Generalized Anxiety among Swiss Health Professions and Non-Health Professions Students: An Open Cohort Study over 14 Months in the COVID-19 Pandemic"

_ijerph, 2021, doi:10.3390/ijerph182010833_

Round 1
Reviewer 1 Report
The article is entitled "Generalized anxiety among Swiss health professions and non-health professions students: a longitudinal open cohort study over 14 months in the COVID-19 pandemic".
It aims to study the evolution of the mental health of HP and non-HP students during the pandemic period.
Overall, the article is of high quality, both in content and form, and meets a need for knowledge on the subject. The literature is mastered and the article is clear.
We do not have many remarks to make about the article except for one. There is a confusion in the article between longitudinal study and repeated cross-sectional study.
Wouldn't it be more appropriate to choose one of the two terms, the one most appropriate to the method? In the method, the author speaks of a repeated cross-sectional study, but otherwise the term longitudinal study appears. However, we do not know whether the students were matched when the data were collected, even partially, and this is not clear in the text and leads to much confusion, both in the interpretation of the results and in the limitations.
Table 1 seems to suggest that the data are not matched, and the variation in the number of people per measurement seems to support this.
This is an important limitation, in that the authors are able to tell if a student participated in multiple measures or not.
Modifications on these elements are needed in the methodology and limitation section of the article.
Reviewer 2 Report
I enjoyed reading your paper. Most of the questions I had while reading were resolved within the paper, probably because your article communicated your findings systematically and in detail. Some limitations appear to be due to the limitations of your data, but the demand of timeliness and importance of the COVID research make your research meaningful enough. I expect this study to contribute to the literature by meeting the need for long-term observational studies of student groups.
As a minor complaint, please consider these points.
- I didn't know which country sample the study was on in the abstract.
- According to Table 1, 22.6% of line 213 appears to be 20.8%, so please check if the minimum value you saw is correct.
- You have performed logistic models for Table 3, and I would like you to show the models in your paper so that the reader can check the specific equations for the models. Also the name of Table 3 must be changed to show your test model. (it was logistic regression tests, not factor analysis or something)
- In your study, the inconsistency in your sample collections produced differences between some variables and others, and forced you to do two different types of regression tests. Would you mind adding this problem to the limitation part of the paper?
I look forward to your further research in the future. Thank you.
Reviewer 3 Report
In the present study authors aimed to 1) examine the course of GAD symptom in HP and non-HP students over 14 months of the COVID-19 pandemic, 2) to compare levels of GAD symptoms among HP students and their peers, and 3) to identify factors associated with GAD symptoms.
This study is well written and brings new insight into mental health in the COVID-19 pandemic.
Materials and Methods and Results sections do not require correction, they are well designed.
Nevertheless, in the introduction or discussion section, it is worth referring to publications on the population of young adults, including medical professions. In these studies, first, a strong correlation between GAD and chronic diseases was demonstrated in patients with autoimmune diseases. In contrast, the second study showed that it is not the status of the profession but the status of the disease that has an impact on mental health.
Wańkowicz, P.; Szylińska, A.; Rotter, I. The Impact of the COVID-19 Pandemic on Psychological Health and Insomnia among People with Chronic Diseases. J. Clin. Med. 2021, 10, 1206. https://doi.org/10.3390/jcm10061206
Wańkowicz, P.; Szylińska, A.; Rotter, I. Insomnia, Anxiety, and Depression Symptoms during the COVID-19 Pandemic May Depend on the Pre-Existent Health Status Rather than the Profession. Brain Sci. 2021, 11, 1001. https://doi.org/10.3390/brainsci11081001
